# Economic evaluation of the cost of different methods of retesting chlamydia positive individuals in England

Katharine J Looker,[1] Erna Buitendam,[2] Sarah C Woodhall,[2] Emma Hollis,[2] Koh-Jun Ong,[2] John M Saunders,[2] Kevin Dunbar,[2] Katherine M E Turner[1]

[1]Population Health Sciences, Bristol Medical School, University of Bristol, Bristol, Bristol, UK
[2]HIV & STI Department, National Infection Service, Public Health England, London, UK

**Correspondence to**
Dr Katharine J Looker;
katharine.looker@bristol.ac.uk

## ABSTRACT

**Objectives** The National Chlamydia Screening Programme (NCSP) in England opportunistically screens eligible individuals for chlamydia infection. Retesting is recommended three 3 months after treatment following a positive test result, but no guidance is given on how local areas should recall individuals for retesting. Here, we compare cost estimates for different recall methods to inform the optimal delivery of retesting programmes.

**Design** Economic evaluation.

**Setting** England.

**Methods** We estimated the cost of chlamydia retesting for each of the six most commonly used recall methods in 2014 based on existing cost estimates of a chlamydia screen. Proportions accepting retesting, opting for retesting by post, returning postal testing kits and retesting positive were informed by 2014 NCSP audit data. Health professionals 'sense-checked' the costs.

**Primary and secondary outcomes** Cost and adjusted cost per chlamydia retest; cost and adjusted cost per chlamydia retest positive.

**Results** We estimated the cost of the chlamydia retest pathway, including treatment/follow-up call, to be between £45 and £70 per completed test. At the lower end, this compared favourably to the cost of a clinic-based screen. Cost per retest positive was £389–£607. After adjusting for incomplete uptake, and non-return of postal kits, the cost rose to £109–£289 per completed test (cost per retest positive: £946–£2,506). The most economical method in terms of adjusted cost per retest was no active recall as gains in retest rates with active recall did not outweigh the higher cost. Nurse-led client contact by phone was particularly uneconomical, as was sending out postal testing kits automatically.

**Conclusions** Retesting without active recall is more economical than more intensive methods such as recalling by phone and automatically sending out postal kits. If sending a short message service (SMS) could be automated, this could be the most economical way of delivering retesting. However, patient choice and local accessibility of services should be taken into consideration in planning.

## Strengths and limitations of this study

► We compared the cost of the chlamydia retest pathway in England across the six most commonly used methods of recalling individuals for retesting, to enable local service planners to assess whether they are delivering retesting economically or should consider an alternative approach.

► Our cost estimates included both clinic retesting and retesting using postal kits.

► We incorporated incomplete uptake, and non-return of postal kits, to estimate cost based on actual patterns of use.

► We did not specifically look at the effect of factors such as gender, country of birth, sexual orientation, perceived risk of infection and presence of symptoms on retest uptake and therefore cost, although no active recall is likely to be associated with similar or higher test positivity than active recall.

► We also did not consider other important factors besides cost such as the demography of the population: for example, automatically sending out postal kits might be the only feasible option in rural areas, and indeed, on-line testing, which was not considered in our analysis, is likely to be the most economical method of all.

## INTRODUCTION

*Chlamydia trachomatis* (chlamydia) is the most commonly diagnosed bacterial sexually transmitted infection (STI) in England.[1] Since 2003, there has been a National Chlamydia Screening Programme (NCSP) in England which opportunistically tests sexually active people aged 15–24 years.[2] NCSP guidelines recommend retesting 3 months after treatment for chlamydia.[3] British Association for Sexual Health and HIV (BASHH) national guidelines recommend retesting people under 25 years of age 3–6 months after treatment.[4] No guidance is given by either the NCSP or BASHH on how local areas should recall individuals for retesting, which can be done in many ways. The 2014 NCSP retesting audit[5] found that the most common methods of recalling individuals for

retesting were as follows: (1) conversation with client at time of test result with no further reminder (32%); (2) reminder card given to client at time of test result with no further reminder (1%); (3) client sent a text message when retest due (29%); (4) client invited by phone call when retest due (8%); (5) testing kit posted to client's chosen address when retest due (5%) and (6) retesting advised at follow-up call with client—text message sent at 3 months (19%). The audit also measured overall retest rates for each recall method, which were calculated from the number that attended a clinic for a retest or returned a postal testing kit, divided by the total number recalled for retesting. This is different to retest uptake, which is the number that attended a clinic for a retest or ordered or were sent a postal testing kit, divided by the total number recalled for retesting, which has cost implications. Retest uptake was not measured by the audit. Unpublished data from the 2017 NCSP retesting audit (Erna Buitendam, personal communication) showed that, for the six most commonly used recall methods in the 2014 audit, retest rates significantly increased for method 1 (client-led) and method 5 (automatic postal test kit) between the 2014 and 2017 audits.

Previous estimates exist for the cost of a clinic-based chlamydia screen.[6 7] However, to our knowledge there are no estimates of the cost of a chlamydia retest, and how this varies by recall method. Specifically, we do not know the best way to balance getting the optimal number of people to retest versus the additional cost of delivering invitations or reminders to retest. Understanding how the cost of retesting varies depending on the approach taken is critical for an optimal programme delivery. Here, we present cost estimates for different recall methods in England, first for the retest pathway itself, and then for the adjusted cost per retest, allowing for incomplete uptake and non-return of postal kits, to impact on cost.

## METHODS

We estimated the cost of chlamydia retesting in England using Microsoft Excel 2016 for each of the six most commonly used recall methods reported in the 2014 NCSP retesting audit[5] (table 1) as follows. First, we entered existing cost estimates for a chlamydia test from Pathway Analytics (costed for clinic-based chlamydia screening for 2011), which excluding a follow-up call was around £45[6] (online supplementary table 1). We used this costing as given. We then added additional costs to reflect costs specifically associated with retesting, such as a nurse-led conversation about retesting after diagnosis, and issuing retest invitations/reminders (eg, by text message [SMS] or phone). A nurse-led conversation about retesting after diagnosis was assumed to involve only extra nurse time to additionally discuss retesting; other associated costs were assumed to be already factored into the cost of a first test. Both an SMS and a phone call were assumed to involve administration time and the cost of the SMS or call itself while the latter was assumed to also include both

nurse time for the actual call and chasing time. In addition, we amended the clinic-based chlamydia test costs to allow for postal testing. Costs were then totalled across the following cost categories: cost of offering retesting, cost of delivering a retest, and cost of processing retest and giving results, as well as overall.

For each of the six recall methods, we costed both the retest pathway, and the adjusted cost per retest (online supplementary figure 1). The adjusted cost per retest accounts for incomplete uptake, and non-return of postal kits, within each cost category. For all methods except method 5 (automatic postal testing kit), we allowed clients to choose either to attend a clinic for retesting or to request a postal testing kit. Thus, for methods 1–4 and method 6, we incorporated the following parameters: retest uptake, the proportion who opt for postal testing, and the return rate of requested kits. Retest uptake for each of the six recall methods was fitted to overall retest rates from the 2014 NCSP audit,[5] taking a value of 24% for the proportion of clients who opt for postal testing (also from the audit), and a value of 67% for the return rate of requested kits.[8] For method 5, uptake was equivalent to overall retest rate and was simply the return rate of postal kits (10%) from the 2014 NCSP audit.[5] Chlamydia retest positivity (12%), which informs the relative weight given to the cost of managing a positive retest result versus managing a negative retest result in the average cost of the chlamydia retesting pathway, was taken from the NCSP audit,[5] and was averaged over all six recall methods due to small numbers by individual method. We also calculated the cost and adjusted cost per retest positive, ie, the cost of finding one positive retest incorporating the cost of other, negative retests, by dividing test costs by the chlamydia positivity. For a table of parameter values, see online supplementary table 2.

The time frame for calculating the parameter values was 10–14 weeks, corresponding to NCSP guidance for retesting. We sense-checked our retesting costs (online supplementary table 1) with health professionals. We conducted two sensitivity analyses. In the first sensitivity analysis, we replaced the parameters for the retesting pathway with those obtained from data for retesting done between 10 and 26 weeks (corresponding to BASHH guidance) (online supplementary table 2). This simply allows more time for clients to retest: there is no additional contact with clients to remind them to retest. In the second sensitivity analysis, we altered staff salary costs from nurse bands to administrator bands for nurse-based costs associated with phone invitations to retest, managing a retest negative and a follow-up call at 3 months for those retesting positive (leaving the nurse-based costs associated with the initial retest conversation and managing a retest positive unchanged). The purpose of this was to show the difference in price that could be achieved if administrative staff instead of nurses contacted clients by phone, except where a lower band of advisor might not be appropriate.

**Table 1** Chlamydia retest costs by recall method

| | Recall method | | | | | |
| --- | --- | --- | --- | --- | --- | --- |
| | 1. Client-led | 2. Reminder card | 3. SMS invitation | 4. Phone invitation | 5. Automatic postal test kit | 6. Advice at follow-up and SMS |
| Number of retest invitations by each method (%), n=2853* (National Chlamydia Screening Programme [NCSP] audit, 2014[5]) | 912 (32%) | 27 (1%) | 840 (29%) | 227 (8%) | 130 (5%) | 528 (19%) |
| Description | Conversation with client at time of test result with no further reminder | Reminder card given to client at time of test result with no further reminder | Client sent text message when retest due | Client invited by phone call when retest due | Testing kit posted to client's chosen address when retest due | Retesting advised at follow-up call with client - text message sent at 3 months |
| **Costs using baseline parameters (10–14 weeks since treatment for first infection)** | | | | | | |
| Cost of chlamydia retesting pathway† | | | | | | |
| Cost of offering retesting | £2.68 | £2.78 | £5.42 | £14.44 | £2.68 | £17.18 |
| Cost of delivering retest | £24.16 | £24.16 | £24.16 | £24.16 | £13.45 | £24.16 |
| Cost of processing retest and giving results | £28.71 | £28.71 | £28.71 | £28.71 | £28.71 | £28.71 |
| TOTAL COST | £55.54 | £55.64 | £58.28 | £67.31 | £44.83 | £70.05 |
| Cost per retest positive | £481 | £482 | £505 | £583 | £389 | £607 |
| Retest uptake (%) | 5 | 4 | 9 | 7 | 10 | 13 |
| Retest rate (%) | 5 | 4 | 8 | 6 | 10 | 12 |
| Adjusted cost per retest incorporating incomplete uptake/ non-return of kits | £109 | £130 | £120 | £289 | £190 | £195 |
| Adjusted cost per retest positive incorporating incomplete uptake/non-return of kits | £946 | £1126 | £1039 | £2506 | £1646 | £1686 |
| **Costs using longer time window for retesting (10–26 weeks since treatment for first infection)** | | | | | | |
| Total cost of chlamydia retesting pathway | £55.38 | £55.48 | £58.12 | £67.15 | £45.32 | £69.89 |
| Cost per retest positive | £344 | £345 | £361 | £417 | £282 | £435 |
| Retest uptake (%) | 16 | 20 | 23 | 18 | 23 | 27 |
| Retest rate (%) | 15 | 19 | 21 | 17 | 23 | 25 |
| Adjusted cost per retest incorporating incomplete uptake/ non-return of kits | £73 | £71 | £82 | £142 | £99 | £126 |
| Adjusted cost per retest positive incorporating incomplete uptake/non-return of kits | £456 | £440 | £508 | £883 | £616 | £780 |
| **Costs if administrators used instead of nurses (10–14 weeks since treatment for first infection)** | | | | | | |
| Total cost of chlamydia retesting pathway | £52.13 | £52.23 | £54.87 | £60.24 | £41.42 | £62.98 |
| Cost per retest positive | £452 | £453 | £476 | £522 | £359 | £546 |
| Adjusted cost per retest incorporating incomplete uptake/ non-return of kits | £106 | £126 | £117 | £227 | £187 | £161 |
| Adjusted cost per retest positive incorporating incomplete uptake/non-return of kits | £917 | £1096 | £1010 | £1963 | £1617 | £1399 |

*Other methods or method not recorded account for the remaining 7% (n=189) of retests.
†Some costs were taken (and some subsequently amended) from the basic cost of a (first) chlamydia test[6] which is under a Creative Commons licence.
© Pathway Analytics.

Since retest rates significantly increased for method 1 (client-led) and method 5 (automatic postal test kit) between the 2014 and (unpublished) 2017 audits (P<0.05), we restricted our analyses to 2014 data only. However, we carried out an analysis of whether retest positivity was statistically significantly different for no active recall (method 1) versus active recall (methods 3 and 6) using both 2014 and 2017 audit data, since there was no statistically significant difference in the positivity rates for each of these methods when comparing 2014 and 2017 data.

### Patient and public involvement

Patients and the public were not involved in this analysis.

### RESULTS

The estimated cost of the chlamydia retest pathway ranged from £45 to £70 per completed test, while the cost per retest positive ranged from £389 to £607 (table 1). Posting testing kits automatically with no further reminder (method 5) was the cheapest recall method, while methods involving inviting clients by phone to retest (methods 4 and 6) were the most expensive. After adjusting for incomplete uptake and non-return of postal kits, the cost per chlamydia retest ranged from £109 to £289 per completed test, while the cost per retest positive ranged from £946 to £2506. Here, the most economical recall method in terms of the adjusted cost per retest was no active recall (method 1). An SMS invitation (method 3) increased retest rates for a comparatively small additional cost. The most expensive methods were still those involving inviting clients by phone to retest (methods 4 and 6). This was despite these methods achieving higher retest rates (6% and 12% for methods 4 and 6, respectively) compared with no active recall (5%). Sending postal testing kits out automatically (method 5) was also an uneconomical way of delivering a retest, due to the cost of non-returned kits. Retest positivity was not statistically significantly different for no active recall (method 1) versus active recall (methods 3 and 6) when 2014 and 2017 audit data were combined.

Extending the retesting period to 10–26 weeks did not impact substantially on the chlamydia retesting pathway cost (range £45–£70) (table 1). However, the adjusted cost per retest incorporating incomplete uptake and non-return of kits was substantially lower (range £71–£126), as was the adjusted cost per retest positive (range £440–£883), than with a tighter retest window, particularly for automatically sending out postal kits (method 5). However, this assumed positivity was higher for the 10–26 week window across all methods. In any case, methods with no or else minimal active recall were still the most economical. Replacing nurse bands with administrator bands only had a substantial impact on costs for those methods where clients were contacted by phone to recall for retest (table 1).

### DISCUSSION

The estimated cost of the chlamydia retest pathway ranged from £45 to £70 per completed test, which at the cheapest end was very similar to the cost of a clinic-based chlamydia screen.[6 7] The cost per retest positive, meanwhile, ranged from £389 to £607. Important differences were seen when uptake and kit return rates were varied. This is because successfully completed retests effectively absorbed the cost of incomplete retests. Here, the most economical recall method involved no active recall after the initial retest conversation. Sending out postal testing kits automatically was an expensive way of doing retesting because of wastage of kits. However, the most expensive methods involved contacting clients by phone to invite them to retest, primarily because of nurse time required.

When the retesting window was increased from 10–14 weeks to 10–26 weeks, all methods of recall had a reduced adjusted cost per retest, due in part to a higher positivity for 10–26 weeks. However, a longer time window means there is further potential for onward transmission, so it is important clients are counselled on the best time to retest.

Active recall increased retest rates, but this did not outweigh the additional cost. We assumed that sending an SMS involved administration time to retrieve clients' details from a database. In our analysis, we considered only the effect of altering staff salary costs from nurse bands to administrator bands for some nurse-based activities. If the time needed to send an SMS could be shortened by automating this process, then an SMS invitation or reminder could be an economical way of increasing retest rates. For example, if the cost of associated administration time is removed, then the adjusted cost per chlamydia retest is £88 and the adjusted cost per retest positive is £765, making sending an SMS the most economical way of delivering retesting. Conversations with health professionals during our study suggested that a shorter administration time to send an SMS was theoretically feasible. We did not find any evidence that retest positivity was different for active recall versus no active recall, meaning there is no evidence that active recall merely results in more negatives being tested. However, evidence from a retesting pilot in South-West England did show that those who retested without being actively recalled had higher chlamydia retest positivity than those who were actively recalled.[8] Furthermore, the unpublished 2017 audit data showed a statistically significant increase in the retest rate for client-led retesting for 10–14 weeks compared with 2014 (Erna Buitendam, personal communication), which could make no active recall even more economical than shown here.

Our analysis was done for the pathway cost of testing for chlamydia alone.[6] Where chlamydia testing is done at the same time as testing for other STIs (such as gonorrhoea), the proportionate cost of testing for chlamydia will be reduced. Another consideration is that since our analysis was carried out, the estimated pathway cost has fallen, which will reduce costs further across all methods

of retesting. However, cost is not the only important factor to consider. For example, no active recall also had the lowest retest rate, although as noted above, active recall may not necessarily identify more infected people if those opting to retest self-select on the basis of their perceived risk or presence of symptoms. We also did not account for the effect on retest uptake of factors such as gender, location of services, country of birth and sexual orientation. The composition of the population is an important consideration in local planning: a large rural population, for example, might affect how retesting needs to be delivered. Given the much higher return rate for requested postal testing kits compared with kits sent out automatically, online testing with automated recall is likely to be the most economical method of all, but was beyond the scope of this analysis.

Our analysis suggests that no active recall after the initial retest conversation is the most economical way of retesting, although an SMS invitation or reminder could be considered. Patient choice and accessibility of services should still be taken into consideration for local delivery planning and it is important that retest uptake is monitored as this determines how economical retesting is.

**Acknowledgements** KJL and KMET thank the National Institute for Health Research (NIHR) Health Protection Research Unit (HPRU) in Evaluation of Interventions at the University of Bristol, in partnership with Public Health England, for research support. We would like to thank Rose Tobin (North East and North Central London Adult Critical Care Network, Royal Free London NHS Foundation Trust), Stephanie Rumsey (East Cheshire NHS Trust), Jan Cremer (Essex Partnership University NHS Foundation Trust) and Stephanie Sawyer (London Borough of Bromley Public Health) for sense-checking the costs, and Pathway Analytics for data on the cost of a chlamydia test.

**Contributors** KJL undertook the itemisation and costing, analysed the results and drafted the manuscript. KMET oversaw the study and provided advice as needed. EB and SCW provided audit data and advised on parameterisation. K-JO helped with sources for costs. KJL, EB, SCW, EH, K-JO, JMS, KD and KMET all contributed to the progress of the study and manuscript revisions.

**Funding** The research was funded by the National Institute for Health Research Health Protection Research Unit (NIHR HPRU) in Evaluation of Interventions at the

University of Bristol in partnership with Public Health England (PHE). The views expressed are those of the author(s) and not necessarily those of the NHS, the NIHR, the Department of Health and Social Care or Public Health England.

**Competing interests** None declared.

**Patient consent for publication** Not required.

**Provenance and peer review** Not commissioned; externally peer reviewed.

**Data sharing statement** There are no additional unpublished data from this study.

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
