## [Reviewer comments · BMJ Open]

ARTICLE DETAILS

TITLE (PROVISIONAL)	An economic evaluation of the cost of different methods of retesting chlamydia positive individuals in England
AUTHORS	Looker, Katharine; Buitendam, Erna; Woodhall, Sarah; Hollis, Emma; Ong, Koh Jun; Saunders, John; Dunbar, Kevin; Turner, Katy

VERSION 1 – REVIEW

REVIEWER	Sarah Creighton Homerton Hospital, London, UK
REVIEW RETURNED	29-Jun-2018

GENERAL COMMENTS	I note the amendments from the previous submission. The following amendments may make the study clearer:- Introduction line 74: (5) omitted before 'testing kit posted to client's....' Methods line 103 'The 2014 NCSP audit[5] measured overall retest rates, which were calculated from the number that attended a clinic for a retest or returned a postal testing kit, divided by the total number recalled for retesting. This is different to retest uptake, which is the number that attended a clinic for a retest or ordered or were sent a postal testing kit, divided by the total number recalled for retesting' The background information, and the rationale for using retest uptake rather than retest rate is better presented in the introduction rather than the methods. Methods line 129 'During the study, we also had access to unpublished data from the 2017 NCSP retesting audit (Erna Buitendam, personal communication). 'et seq. This background information may be better presented in the introduction, along with an explicit recognition that exclusion 14% of the retests could skew the data. Table 1. I note that row 6 refers to the retest rate (not the retest uptake as per the text). Is this intentional? I note the positivity rate is assumed to be consistent across all method types. It is therefore difficult to explain why the ratio between Retest rate*Adjusted cost per retest incorporating incomplete uptake/non-return of kits/Adjusted cost per retest positive incorporating incomplete uptake/non-return of kits is not consistent across all method types and the calculation of this should be explained. REsults line 164. Is ther any evidence o the administration time for sending an SMS being either five or three minutes?
---

REVIEWER	Kirsty Smith The Kirby Institute, UNSW Sydney, Australia
REVIEW RETURNED	13-Jul-2018

GENERAL COMMENTS	The authors have improved the paper by including the cost per positive retest however these new results are not considered in the abstract and discussion. A major limitation of the paper from a service provider perspective is that it focuses on the cost of each retesting method without adequately contextualising this with regard to the ultimate goals of uptake and retest positivity. For example, as would be expected, no active recall was the cheapest method but the uptake was among the lowest at only 5%, whereas postal test kits (PTKs) were among the more expensive options when taking into account the cost of non-returned kits, but the uptake was second highest at 10%. Therefore the discussion should be modified further to reflect the new data and a more nuanced interpretation of the results. The authors have noted that it was not possible to include the retest positivity rate by method due to small numbers however it would be helpful to include more detail regarding the methods for calculating the cost and adjusted cost per positive retest. It is good to see that the authors have included the following comment in the discussion: Lines 188-190: 'A further limitation is we did not account for the effect on retest uptake of different demographic and clinical factors, such as gender, location of services, country of birth, sexual orientation and presence of symptoms.' However the implications of this should be made clearer i.e. that it makes it difficult to fairly compare the uptake (and therefore the cost) of the different methods. Further, while automated postal test kits (PTKs) were deemed uneconomical due to the cost of unreturned kits, the estimated return rate for requested kits is much higher (67%). This warrants further discussion. And finally, although the cost of SMS reminders was marginally cheaper when admin staff were used instead of nurses, it should also be noted that the cost could be significantly reduced if SMS messages were automated.
--

VERSION 1 – AUTHOR RESPONSE

Reviewer(s)' Comments to Author:

Reviewer: 1

I note the amendments from the previous submission. The following amendments may make the study clearer:-

Introduction line 74: (5) omitted before 'testing kit posted to client's....' Thank you for spotting this! We have now inserted the missing "(5)".

Methods line 103 'The 2014 NCSP audit[5] measured overall retest rates, which were calculated from the number that attended a clinic for a retest or returned a postal testing kit, divided by the total number recalled for retesting. This is different to retest uptake, which is the number that attended a clinic for a retest or ordered or were sent a postal testing kit, divided by the total number recalled for retesting' The background information, and the rationale for using retest uptake rather than retest rate is better presented in the introduction rather than the methods.

The paragraph in question (slightly amended from the above) has been moved to the Introduction.

Methods line 129 'During the study, we also had access to unpublished data from the 2017 NCSP retesting audit (Erna Buitendam, personal communication). 'et seq. This background information may be better presented in the introduction, along with an explicit recognition that exclusion [of] 14% of the retests could skew the data.

We have moved (and slightly amended) part of the paragraph in question to the Introduction. We are unclear what the Reviewer is referring to re exclusion [of] 14% of the retests. Does the Reviewer mean the 7% where the method of recall was other or not recorded? We would expect a similar percentage in the 2017 NCSP data – and note the percentages would not simply sum.

Table 1. I note that row 6 refers to the retest rate (not the retest uptake as per the text). Is this intentional?

Yes this is retest rate (from the audit), not retest uptake. We have added rows to Table 1 for retest uptake, for clarity.

I note the positivity rate is assumed to be consistent across all method types. It is therefore difficult to explain why the ratio between Retest rate*Adjusted cost per retest incorporating incomplete uptake/non-return of kits/Adjusted cost per retest positive incorporating incomplete uptake/nonreturn of kits is not consistent across all method types and the calculation of this should be explained.

This ratio is consistent but is not exact using the numbers in Table 1 due to rounding.

Results line 164. Is there any evidence of the administration time for sending an SMS being either five or three minutes?

Yes, from speaking to the health professionals involved in our study. We have inserted the following sentence to the Discussion to reflect this: "Conversations with health professionals during the course of our study suggested that a shorter administration time to send an SMS was theoretically feasible."

Reviewer: 2

1. The authors have improved the paper by including the cost per positive retest however these new results are not considered in the abstract and discussion. These results are now mentioned in both the abstract and discussion.

2. A major limitation of the paper from a service provider perspective is that it focuses on the cost of each retesting method without adequately contextualising this with regard to the ultimate goals of uptake and retest positivity. For example, as would be expected, no active recall was the cheapest method but the uptake was among the lowest at only 5%, whereas postal test kits (PTKs) were among the more expensive options when taking into account the cost of non-returned kits, but the

uptake was second highest at 10%. Therefore the discussion should be modified further to reflect the new data and a more nuanced interpretation of the results.

We have added the following sentence to the Strengths and limitations section: “We did not specifically look at the effect of clinical factors on cost, although no active recall is likely to be associated with similar or higher test positivity than active recall.”

We have also amended the Discussion as follows: “However, cost is not the only important factor to consider. For example, the cheapest recall methods also had the lowest retest rates, although as noted above, active recall may not necessarily identify more infected people if those opting to retest self-select on the basis of their perceived risk or presence of symptoms.”

3. The authors have noted that it was not possible to include the retest positivity rate by method due to small numbers however it would be helpful to include more detail regarding the methods for calculating the cost and adjusted cost per positive retest.

We have added additional details to the Methods section on how we calculated the cost estimates.

4. It is good to see that the authors have included the following comment in the discussion: Lines 188-190: ‘A further limitation is we did not account for the effect on retest uptake of different demographic and clinical factors, such as gender, location of services, country of birth, sexual orientation and presence of symptoms.’ However the implications of this should be made clearer i.e. that it makes it difficult to fairly compare the uptake (and therefore the cost) of the different methods. Further, while automated postal test kits (PTKs) were deemed uneconomical due to the cost of unreturned kits, the estimated return rate for requested kits is much higher (67%). This warrants further discussion.

We have added the following sentence to the Strengths and limitations section: “We also did not consider other important factors besides cost such as the demography of the population: for example, automatically sending out postal kits might be the only feasible option in rural areas, and indeed, on-line testing, which was not considered in our analysis, is likely to be the most economical method of all.”

We have also amended the Discussion as follows: “We also did not account for the effect on retest uptake of factors such as gender, location of services, country of birth, and sexual orientation. The composition of the population is an important consideration in local planning: a large rural population, for example, might affect how retesting needs to be delivered. Given the much higher return rate for requested postal testing kits compared to kits sent out automatically, online testing with automated recall is likely to be the most economical method of all, but was beyond the scope of this analysis.”

And finally, although the cost of SMS reminders was marginally cheaper when admin staff were used instead of nurses, it should also be noted that the cost could be significantly reduced if SMS messages were automated.

We have amended the Discussion as follows: “We assumed that sending an SMS involved administration time to retrieve clients’ details from a database. In our analysis we considered only the effect of altering staff salary costs from nurse bands to administrator bands for some nursebased activities. If the time needed to send an SMS could be shortened by automating this process, then an SMS invitation or reminder could be an economical way of increasing retest rates.”

VERSION 2 – REVIEW

REVIEWER	Sarah Creighton Homerton Hospital, UK
REVIEW RETURNED	17-Oct-2018

GENERAL COMMENTS	It is not immediately apparent as to why the cost of offering retesting for 'Conversation with client at time of test' is £2.68, for 'Client sent text message when retest due' is £5.42 yet "Conversation with client at time of test + sent text message when retest due' is £17.18 rather than £8.10. All subsequent calculations depend on this The abstract and the discussion make reference to the fact that the costs of SMS reminders can be reduced if SMS texting is automated. Can this be quantified? The 'however' at the end of line 189 '(An SMS invitation (method three) increased retest rates for comparatively small additional cost, however.)' feels redundant
--

REVIEWER	Kirsty Smith Kirby Institute, UNSW Sydney, Australia
REVIEW RETURNED	29-Oct-2018

GENERAL COMMENTS	An economic evaluation of the cost of different methods of retesting chlamydia positive individuals in England Amendments to the previous version are noted. Please find some further suggestions below. Strengths and limitations Lines 59-60: We did not specifically look at the effect of clinical factors on cost, although no active recall is likely to be associated with similar or higher test positivity than active recall. It is unclear what clinical factors the authors are referring to however it is important to note that the effect of factors such as gender, type and location of services, country of birth and sexual orientation, on retest uptake and therefore cost, were not considered in this evaluation. Discussion: Lines 201-203 For example, the cheapest recall methods also had the lowest retest rates This statement is not strictly true as SMS reminders were among the cheapest methods but had higher rates of retesting than recall method 4- phone invitation, which was one of the two most expensive strategies. However it should be clearly noted that while no active recall was the cheapest method, the uptake was lowest (5%). Table 1 Line 43- it should be retest uptake rather than update Heading- Costs if administrators used instead of nurses- for clarity please add in brackets (10-14 weeks since treatment for first infection)
---

VERSION 2 – AUTHOR RESPONSE

Reviewer(s)' Comments to Author:

Reviewer: 1

1. It is not immediately apparent as to why the cost of offering retesting for 'Conversation with client at time of test' is £2.68, for 'Client sent text message when retest due' is £5.42 yet "Conversation with client at time of test + sent text message when retest due' is £17.18 rather than £8.10. All subsequent calculations depend on this.

We have amended this Method section to better explain this, as follows: “We then added additional costs to reflect costs specifically associated with retesting, such as a nurse-led conversation about retesting after diagnosis, and issuing retest invitations/reminders (e.g., by text message [SMS] or phone). A nurse-led conversation about retesting after diagnosis was assumed to involve only extra nurse time to additionally discuss retesting; other associated costs were assumed to be already factored into the cost of a first test. Both an SMS and a phone call were assumed to involve administration time and the cost of the SMS or call itself, while the latter was assumed to also include both nurse time for the actual call as well as chasing time.”

2. The abstract and the discussion make reference to the fact that the costs of SMS reminders can be reduced if SMS texting is automated. Can this be quantified?

The following sentence has been added to the Discussion, “For example, if the cost of associated administration time is removed, then the adjusted cost per chlamydia retest is £88 and the cost per retest positive is £765, making sending an SMS the most economical way of delivering retesting.”

3. The 'however' at the end of line 189 '(An SMS invitation (method three) increased retest rates for comparatively small additional cost, however.)' feels redundant

This has now been removed.

Reviewer: 2

Amendments to the previous version are noted. Please find some further suggestions below.

Strengths and limitations

1. Lines 59-60: We did not specifically look at the effect of clinical factors on cost, although no active recall is likely to be associated with similar or higher test positivity than active recall. It is unclear what

clinical factors the authors are referring to however it is important to note that the effect of factors such as gender, type and location of services, country of birth and sexual orientation, on retest uptake and therefore cost, were not considered in this evaluation.

We have amended this sentence to, “We did not specifically look at the effect of factors such as gender, country of birth, sexual orientation, perceived risk of infection and presence of symptoms on retest uptake and therefore cost, although no active recall is likely to be associated with similar or higher test positivity than active recall.”

Discussion:

1. Lines 201-203 For example, the cheapest recall methods also had the lowest retest rates. This statement is not strictly true as SMS reminders were among the cheapest methods but had higher rates of retesting than recall method 4- phone invitation, which was one of the two most expensive strategies. However it should be clearly noted that while no active recall was the cheapest method, the uptake was lowest (5%).

This is noted. We have amended this to, “For example, no active recall also had the lowest retest rate...”

Table 1

1. Line 43- it should be retest uptake rather than update

Thank you for spotting this! This has now been corrected.

2. Heading- Costs if administrators used instead of nurses- for clarity please add in brackets (10-14 weeks since treatment for first infection)

This has been done.